# Biosynthesis of ansamitocin P-3 incurs stress on the producing strain *Actinosynnema pretiosum* at multiple targets

Qungang Huang[1,2], Xin Zhang[1,2], Ziyue Guo[1,2], Xinnan Fu[1,2], Yilei Zhao [1,2], Qianjin Kang [1,2,3✉] & Linquan Bai [1,2,3✉]

Microbial bioactive natural products mediate ecologically beneficial functions to the producing strains, and have been widely used in clinic and agriculture with clearly defined targets and underlying mechanisms. However, the physiological effects of their biosynthesis on the producing strains remain largely unknown. The antitumor ansamitocin P-3 (AP-3), produced by *Actinosynnema pretiosum* ATCC 31280, was found to repress the growth of the producing strain at high concentration and target the FtsZ protein involved in cell division. Previous work suggested the presence of additional cryptic targets of AP-3 in ATCC 31280. Herein we use chemoproteomic approach with an AP-3-derived photoaffinity probe to profile the proteome-wide interactions of AP-3. AP-3 exhibits specific bindings to the seemingly unrelated deoxythymidine diphosphate glucose-4,6-dehydratase, aldehyde dehydrogenase, and flavin-dependent thymidylate synthase, which are involved in cell wall assembly, central carbon metabolism and nucleotide biosynthesis, respectively. AP-3 functions as a non-competitive inhibitor of all three above target proteins, generating physiological stress on the producing strain through interfering diverse metabolic pathways. Overexpression of these target proteins increases strain biomass and markedly boosts AP-3 titers. This finding demonstrates that identification and engineering of cryptic targets of bioactive natural products can lead to in-depth understanding of microbial physiology and improved product titers.

[1] State Key Laboratory of Microbial Metabolism, Shanghai-Islamabad-Belgrade Joint Innovation Center on Antibacterial Resistances, School of Life Sciences and Biotechnology, Shanghai Jiao Tong University, Shanghai 200240, China. [2] Joint International Research Laboratory of Metabolic and Developmental Sciences, Shanghai Jiao Tong University, Shanghai 200240, China. [3]These authors jointly supervised this work: Qianjin Kang, Linquan Bai. ✉email: qjkang@sjtu.edu.cn; bailq@sjtu.edu.cn

Microbes, exemplified by actinobacteria and fungi, are rich repertoire of natural products with diverse structures and bioactivities, which are used as antibacterial, antifungal, antitumor, immunosuppressive, cholesterol-lowering agents, and so on[1]. Due to the wide applications of microbial natural products in clinic and agriculture, their modes of action and targeted cellular components have been well characterized in diverse pathogens and infected tissues. For example, the antibacterial penicillin targets transpeptidases, belonging to the penicillin-binding proteins (PBPs), and inhibits the peptidoglycan biosynthesis in bacteria[2]; The immunosuppressant rapamycin binds to the human FK506 binding protein FKBP12, and subsequently the formed binary complex inhibits the target of rapamycin (TOR) protein to exert the immunosuppressive effect[3].

Interestingly, owing to the development of chemoproteomic technology using drug-derived chemical probes, more drug binding proteins involved in multiple physiological pathways have been identified, implying additional biological functions of a designated drug[4]. Using synthetic β-lactam derivatives with different side chain substitutions as probes for in vivo labeling, Staub and Sieber identified not only the known PBPs, but also non-PBP bacterial targets, including the virulence factor ClpP and a resistance-related β-lactamase[5]. Recently, Sun et al. identified STAT3 as a new target of rapamycin and its suppression of tumor growth, using a photoactive rapamycin analog[6].

However, the effects of natural product production upon their producing hosts have been underexplored. The cellular targets of a few anti-infectives were largely deduced from the known target proteins in pathogens, and corresponding resistant mechanisms were characterized in the producing strains[7]. As the target of the anti-tuberculosis rifamycin is the β-subunit of bacterial RNA polymerase[8], the counterpart of which with mutations of N447,

D438, and Q432 was identified in the rifamycin-producing strain *Amycolatopsis mediterranei* S699; and these mutants confer self-protection to the host[9]. Similarly, a designated 23S rRNA *N*-methyltransferase gene *lmrB* was found to be present in the lincomycin biosynthetic gene cluster of *Streptomyces lincolnensis* to provide resistance to supplemented lincomycin[10].

Ansamitocins (Fig. 1a) are actinobacteria-produced maytansinoids with extraordinary antitumor potency; they specifically target microtubules and have served as the "warheads" in immunoconjugates for the treatment of various types of metastatic breast cancers. Ansamitocins are produced by *Actinosynnema*, *Amycolatopsis* and *Nocardiopsis*, among which the members of the genus *Actinosynnema* are mainly used for industrial fermentation of ansamitocins[11–13]. Recently, we found that ansamitocin P-3 (AP-3) at a high concentration suppressed the growth of the producing strain, *Actinosynnema pretiosum* ATCC 31280 (hereafter as ATCC 31280) (Supplementary Table 1), by targeting the cell division protein FtsZ, a β-tubulin-like protein in bacteria. Overexpression of the FtsZ-encoding gene obviously improved the resistance of the recombinant strain to AP-3, as evidenced by a better growth than the starting strain in the presence of 300 mg/L AP-3. However, growth of the recombinant strain still suffered inferior biomass in the presence of exogenous AP-3, suggesting that FtsZ might not be the only target of AP-3[14]. Therefore, we proposed that other cryptic target(s) of AP-3 might be present in the producing strain, and identification of these targets would be crucial to understand the stress caused by AP-3 biosynthesis and then further improve the AP-3 titer.

Thus, we used a specific photoaffinity AP-3 derived probe, combined with mass spectrometry (MS)-based quantitative analysis, to screen the cryptic targets of AP-3 in the producing strain ATCC 31280. Our analyses revealed that the biosynthesis of AP-3

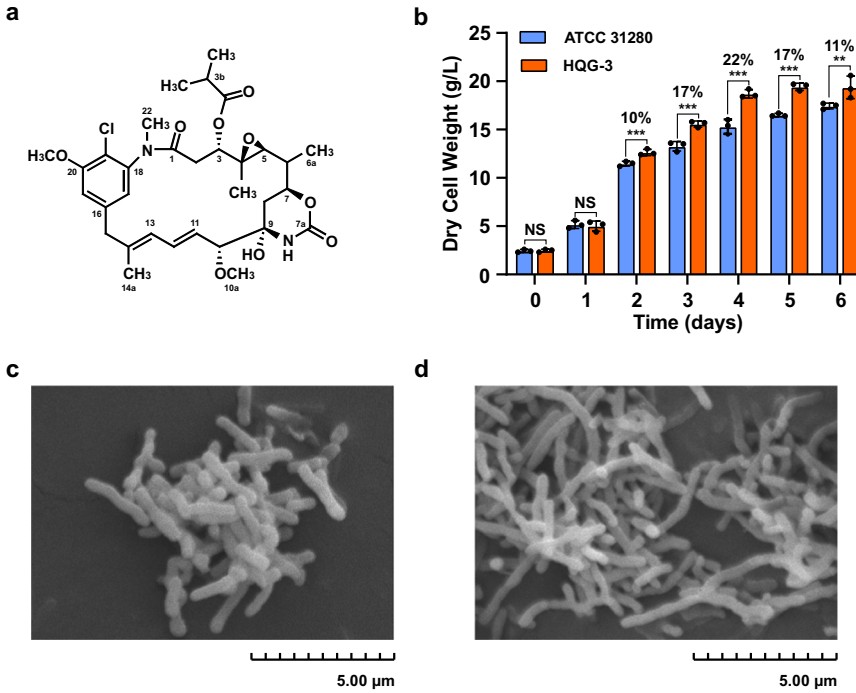

**Fig. 1 AP-3 biosynthesis incurred stress on growth and morphology of *A. pretiosum* ATCC 31280. a** Chemical structure of AP-3. Key numbering is labeled in the structure. **b** Growth profiles of wild-type ATCC 31280 and AP-3-null mutant HQG-3 in YMS medium. The discrepancy of dry cell weight between ATCC 31280 and HQG-3 is shown as percentage numbers. Data are presented as mean values±S.D. from three independent experiments. *P* values were calculated using a two-tailed Student's *t*-test assuming unequal variance; *P < 0.1; **P < 0.05; ***P < 0.01. **c** Mycelia of strain ATCC 31280 in stationary phase showing short rods after fragmentation. **d** Branched and partly fractured mycelia of HQG-3 in stationary phase. Total length of the scale bar of the SEM image is 5 μm.

interfered the physiology of ATCC 31280 in a multi-target manner, with deoxythymidine diphosphate glucose-4,6 dehydratase (dTGD), aldehyde dehydrogenase (ALDH), and flavin-dependent thymidylate synthase (FDTS) as AP-3 targets. The recombinant strains that overexpressed these proteins exhibited increased biomass under AP-3 exposure and also remarkably higher titers of AP-3.

## Results

**Chemoproteomics and MS-based identification of cryptic targets of AP-3 in producing strain ATCC 31280.** To explore the potential ability of AP-3 (Fig. 1a) to cause physiological perturbations of the producing strain, we disrupted the whole ansamitocin biosynthetic gene cluster (*ansa*-cluster) through double crossover recombination in strain ATCC 31280 using pJTU1278-derived plasmid (Supplementary Table 2). This was achieved by cloning PCR-amplified upstream and downstream flanking sequences with two pairs of designated primers (Supplementary Table 3), generating the recombinant strain HQG-3 as a reference strain (Supplementary Fig. 1). The growth of the two strains was compared in YMS medium (Supplementary Table 4), and the results demonstrated that the biomass of HQG-3 was superior to that of strain ATCC 31280. The differences in biomass gradually increased during exponential growth, so that the biomass of HQG-3 was 22% greater than that of the wild-type strain by the fourth day of fermentation (Fig. 1b), likely due to the absence of AP-3 production in HQG-3. Observations on mycelial morphology in stationary phase by scanning electron microscopy revealed that strain ATCC 31280 had a morphology of short rods following fragmentation[15], while the mycelia of HQG-3 were branched and partially fractured (Fig. 1c, d). These results suggested that the biosynthesis of AP-3 at its low concentration also incurs physiological stress, causing the inferior growth of the producing strain.

To identify potential binding targets of AP-3 within strain ATCC 31280, we designed a specific chemoproteomic probe and performed MS-based quantitative analysis. According to the structure-activity relationship of AP-3, the methylation at C20 has a negligible contribution to its biological activity[13]. Consequently, we reasoned that this methyl group could be replaced by a diazirine-alkyne moiety (YNE) to generate a probe. Therefore, we disrupted the methyltransferase-encoding gene *asm7*[16] in strain ATCC 31280 to generate mutant HQG-1 (Supplementary Fig. 2a, b), which accumulated 20-demethyl-ansamitocin P-3 (QG-1), as determined by liquid chromatography (LC)-MS analysis (Supplementary Fig. 2c–f). The YNE moiety was installed onto the phenolic hydroxyl group at C20 of QG-1 via the application of $K_2CO_3$ to generate the photoaffinity probe QG-YNE (Fig. 2a). The chemical structure of compound QG-YNE was confirmed by MS and nuclear magnetic resonance analysis (Supplementary Figs. 3–5). Bioactivity analysis using the AP-3 sensitive yeast *Filobasidium uniguttulatum* as the indicator strain[17] demonstrated that probe QG-YNE inhibited the yeast growth at a comparable level to AP-3 (Fig. 2b), indicating that the YNE moiety attachment did not interfere with the biological potency of QG-YNE.

To identify the AP-3 binding proteins, cell lysates of strain ATCC 31280 were pre-incubated with QG-YNE for 2 h and then irradiated with UV light. Following a click reaction with streptavidin magnetic beads, we carried out pull-down

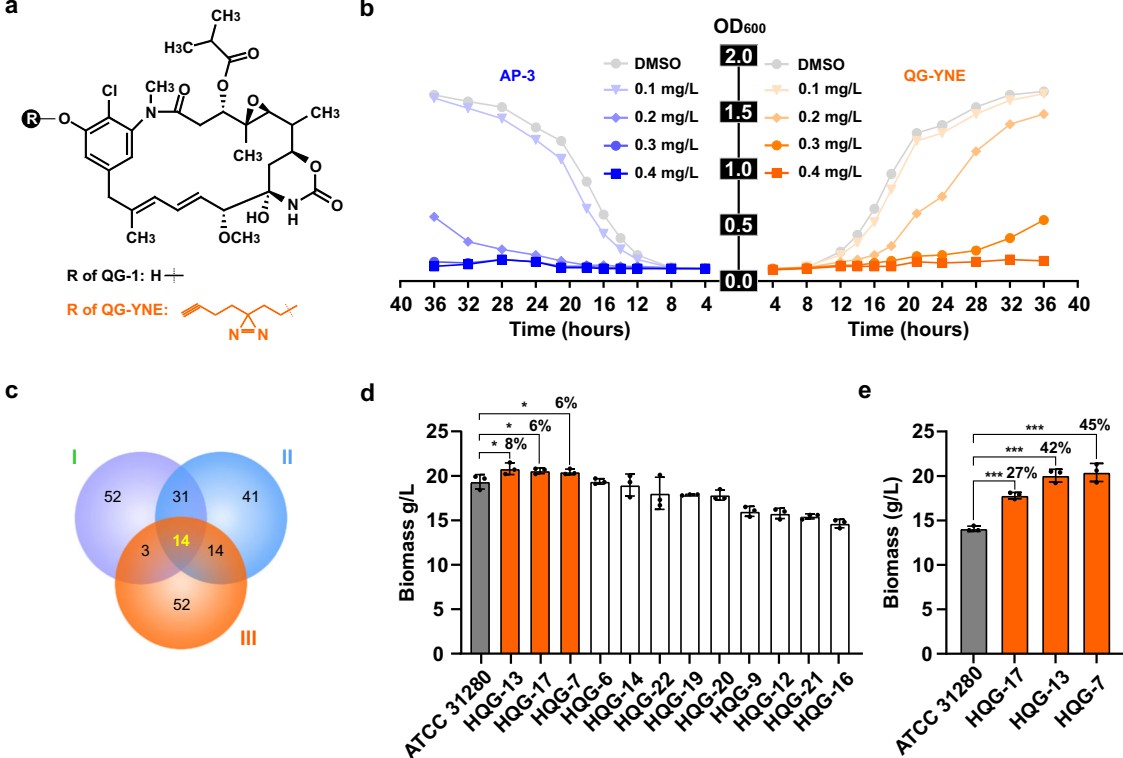

**Fig. 2 Chemoproteomic profiling identified targets of ansamitocin P-3 in strain ATCC 31280. a** Chemical structures of QG-1 and QG-YNE. **b** Comparison of the bioactivities of AP-3 and probe QG-YNE against a yeast indicator strain. **c** Venn diagram showing an overlap of 14 AP-3 binding proteins in the high-abundance fractions from three chemoproteomic assays with different UV exposure times. **d** Tolerance of recombinant strains overexpressing AP-3 binding proteins to exogenous AP-3 (200 mg/L). The discrepancy of biomass between ATCC 31280 and recombinant strains are shown as percentage numbers. Orange bars, biomass higher than that of the wild-type; white bars, biomass lower than that of the wild-type; grey bar, wild-type. *$P < 0.1$; **$P < 0.05$, ***$P < 0.01$. **e** Resistance of HQG-7, HQG-13 and HGQ-17 to 400 mg/L of exogenous AP-3.

manipulations to capture and enrich proteins bound to QG-YNE. Beads were separated from the proteins by in situ tryptic digestion, and the reactions were then labeled with isobaric mass-tagging reagents (TMT-126/127) containing unique reporters (Supplementary Fig. 6). Finally, peptide fragments were analyzed by MS, and the obtained functional proteomic information was filtered by a factor >1 for the ratio of TMT-127/126.

The recovered proteins (fragments) were compared with the annotated proteins of strain ATCC 31280, and from three independent surveys of AP-3 binding proteins using gradually reduced UV exposure times, we obtained enrichment of 708, 490 and 83 candidates, respectively. The effects of varied UV irradiations on individual outcomes were reproducible, and Venn diagram analysis revealed 14 binding proteins that were common to the three UV exposure times (Fig. 2c). In addition to methyltransferase Asm7, which methylates the ansamitocin biosynthetic intermediate[16], the other 13 candidate proteins were associated with diverse metabolic pathways, including aldehyde metabolism, cell wall synthesis, nucleic acid biosynthesis, and regulatory pathways (Supplementary Table 5). These results suggested that the stress on growth and morphological changes brought by AP-3 biosynthesis could arise through different metabolic pathways.

**AP-3 targeted different pathways to interfere with the growth of the producing strain.** As most of the AP-3 binding proteins participate in essential metabolic processes, such as cell division, cell-wall assembly, nucleotide biosynthesis and central carbon metabolism, it was not possible to use gene disruption to examine their possible impacts on the producing strain. Therefore, we designed a set of integrative overexpression plasmids containing each of the binding protein genes cloned under the control of a strong constitutive promoter $kasO$p$^{*}$[18] (Supplementary Figs. 7, 8). Asm7 in this case was excluded as it is responsible for the methylation at C20 position of AP-3, and *APASM_6814* and *APASM_6815* form a gene cassette for their co-expression. Therefore, only 12 overexpression plasmids were constructed and introduced separately into strain ATCC 31280 (Supplementary Table 2). The resulting recombinant strains were assessed for their tolerance to AP-3 by supplementing the fermentation cultures with a final concentration of 200 mg/L AP-3. The biomass was then evaluated on the fifth day, when the cultures reached stationary phase. Strains HQG-7, HQG-13 and HQG-17, which overexpressed aldehyde dehydrogenase (ALDH) (coded by gene *APASM_1052*), flavin-dependent thymidylate synthase (FDTS) (coded by gene *APASM_5765*) and deoxythymidine diphosphate glucose-4,6 dehydratase (dTGD) (coded by gene *APASM_6307*), respectively, exhibited better AP-3 tolerance than the wild-type strain among all 12 mutants carrying overexpression plasmids (Fig. 2d). Strains HQG-7, HQG-13 and HQG-17 were further challenged using a higher concentration of AP-3 (400 mg/L), and quantitative analysis of the biomass on the fifth day of fermentation revealed better growth of these strains, with significantly enhanced tolerance of AP-3 as determined by boosting of the biomass by 45%, 42% and 27%, respectively, in comparison to that of ATCC 31280 (Fig. 2e). As deduced from the annotation and functional studies in other bacteria on these three proteins, *APASM_1052*-coded ALDH catalyzes the oxidation of aldehydes; *APASM_5765*-coded FDTS participates in the synthesis of dTMP from dUMP; and *APASM_6307*-coded dTGD mediates dTDP-L-rhamnose formation involved in the synthesis of bacterial cell wall and capsule (Supplementary Fig. 9)[19], indicating the diverse metabolic processes impacted by AP-3.

**AP-3 inhibited the catalytic ability of dTGD to supply dTDP-L-rhamnose for cell wall assembly.** dTGD oxidizes the C4 hydroxyl group of D-glucose with NAD(H) as a cofactor, followed by dehydration, which results in the generation of the dTDP-L-rhamnose precursor dTDP-6-deoxy-D-*xylo*-4-hexulose (Supplementary Fig. 10). Various genetic studies have determined that abrogation of the dTDP-L-rhamnose biosynthesis greatly attenuates the virulence, and can even influence the viability, of some pathogenic bacteria[20]. To examine the interaction between AP-3 and dTGD, we purified heterologously expressed dTGD from *E. coli* (Supplementary Fig. 11) and used surface plasmon resonance (SPR) to analyze the dTGD and AP-3 interaction at varied concentrations. AP-3 was determined to bind to dTGD with an affinity constant $K_D$ of $3.60 \times 10^{-4}$ M, which was similar to the $K_D$ for the interaction between Asm7 from the AP-3 biosynthetic pathway[16] and AP-3 (Fig. 3a, Supplementary Table 6). MS analysis of the dTGD peptide fragments bound to the photoaffinity probe QG-YNE revealed that peptide 314-DNRDWWEPLKQR-325 was the specific incorporating peptide, which had the expected increase of 712.31 Da (equivalent to QG-YNE with a loss of $N_2$) in molecular weight, relative to the unmodified peptide fragment. Further analysis of the secondary MS (MS2) spectrum indicated that the labeling located to the side chain carboxyl groups of Arg325 and Gln324 of dTGD (Fig. 3b, Supplementary Fig. 12).

To gain a full understanding of the AP-3 interference with the catalytic ability of dTGD, the molecular architecture of dTGD was established with the AlphaFold2 server. The flexible structure alignment of homologous proteins and DesIV from *Streptomyces venezuelae* (PDB entry 1r66) displayed that their architectures were highly similar with a RMSD of 0.41 and the conserved catalytic diad of Asp128 and Glu129 (Supplementary Fig. 13). Molecular docking analysis for AP-3 and dTGD suggested that AP-3 bound to the targeted protein with an affinity of −5.96 kcal/mol, through hydrogen-bond and hydrophobic interactions with adjacent amino acid residues in the binding region. The binding area of AP-3 was away from the catalytic active site of dTGD, and the distances between the hydroxyl group at C20 of AP-3 and Asp128 and Glu129 of dTGD were 34.40 Å and 30.50 Å, respectively (Fig. 3c). The hydroxyl group at C20, which was modified to generate the photoaffinity probe QG-YNE, was near the Arg325 and Gln324 residues of dTGD, at a distance appropriate for the photoaffinity group to crosslink with the residues, consistent with the MS2 analysis (Supplementary Fig. 14).

Furthermore, we performed a quantitative analysis of dTGD activity using different concentrations of dTDP-D-glucose as substrate and NAD$^+$ as cofactor, as monitored by microplate reader at 320 nm ($A_{320}$) (Supplementary Fig. 15). Under optimized conditions, the $k_{cat}$ and $K_m$ of dTGD were measured as 43.29 min$^{-1}$ and 0.15 mM, respectively. The $k_{cat}$ and $K_m$ values were also determined in the presence of 0.31 mM AP-3 as 34.46 min$^{-1}$ and 0.17 mM, respectively, or in the presence of 0.63 mM AP-3 as 29.38 min$^{-1}$ and 0.18 mM, respectively (Fig. 3d). The inhibitory constant ($K_I$) of AP-3 was calculated as 1.10 mM based on the associations between $K_m$ values and different concentrations of AP-3 (Supplementary Table 7). These results suggested that AP-3 serves as a non-competitive inhibitor of dTGD.

As disruption of the gene encoding dTDP-D-glucose 4,6-dehydratase resulted in the elongation of cells by up to 40-fold in *Proteus mirabilis*[21], we compared the mycelial morphology of strains ATCC 31280 and HQG-17. Interestingly, the average single-cell length of HQG-17 was about 1.65 times longer than that of ATCC 31280 statistically as observed by scanning electron

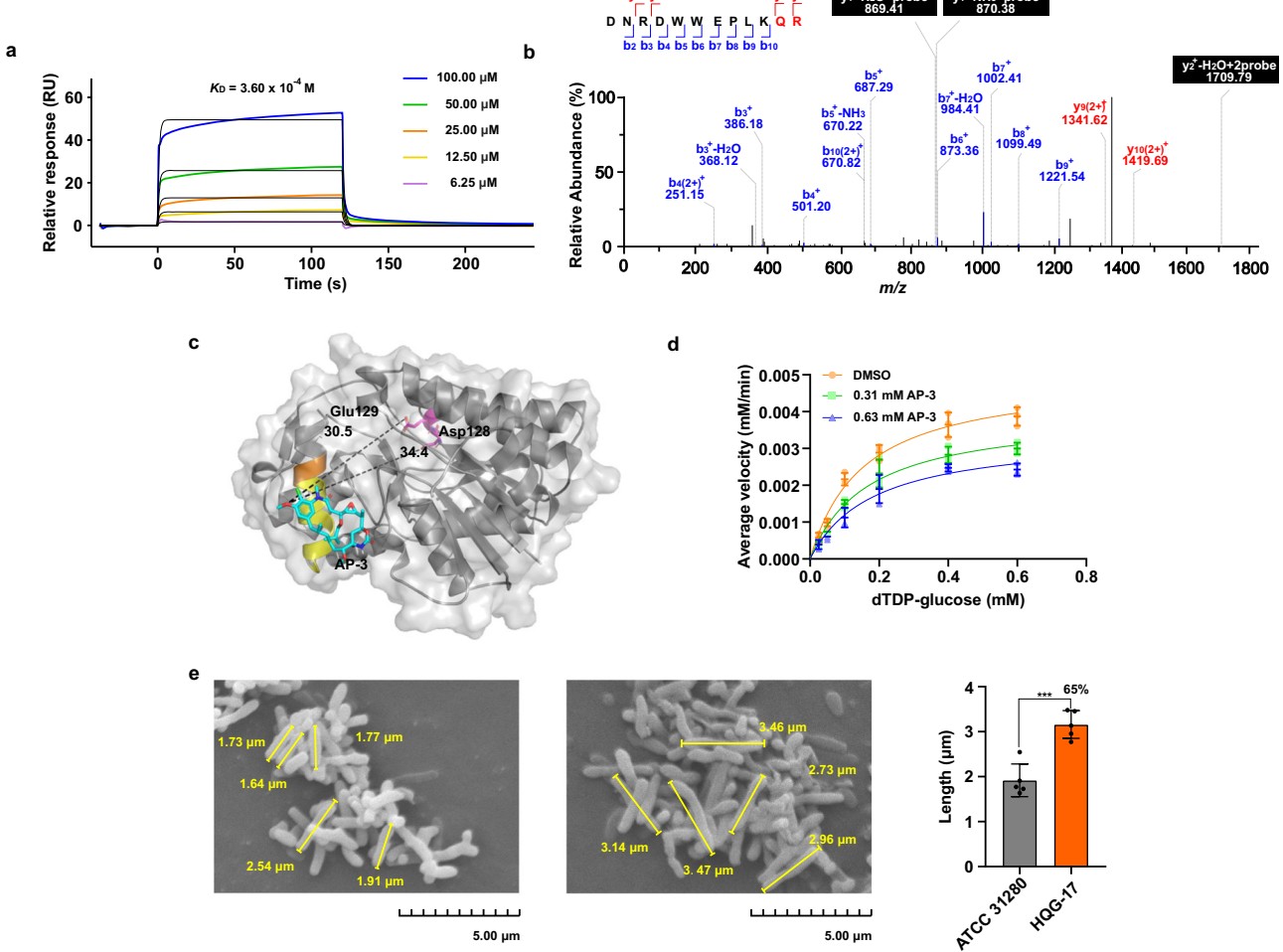

**Fig. 3 dTGD involved in dTDP-L-rhamnose supply is a direct target of AP-3. a** The interaction between AP-3 and dTGD was detected by SPR, giving a $K_D$ value of $3.60 \times 10^{-4}$ M. **b** MS2 spectrum of a QG-YNE-modified peptide from dTGD (314-DNRDWWEPLKQR-325 (Q324 + 712.31 Da) (R + 712.31 Da)). **c** Overall view of the AP-3 and dTGD homo-monomer complex. The distances (Å) between the conserved catalytic amino acids and AP-3 are indicated by dashed lines. Light grey, dTGD homo-monomer; sky blue, AP-3; yellow, peptide labeled by QG-YNE; orange, amino acid modified by QG-YNE; violet, conserved amino acid of dTGD. **d** Inhibitory kinetics of AP-3 with dTGD are displayed by double reciprocal plots obtained using 0.31 or 0.63 mM AP-3 or 1% (v/v) DMSO (the control). Data are presented as mean values with error bars representing S.D. from three independent experiments. **e** Mycelial morphology of strain ATCC 31280 (left panel) and HQG-17 (middle panel) on the fifth day of fermentation. Right panel, comparison of mycelial rod lengths. The discrepancy of mycelium length between ATCC 31280 and HQG-17 is shown as percentage numbers. Total length of the scale bar of the SEM image is 5 µm. Data are presented as mean values with error bars representing S.D. from five independent experiments; $^{***}P < 0.01$.

microscopy with stationary phase cultures (Fig. 3e), suggesting that AP-3 might inhibit dTDP-L-rhamnose supply in the producing strain by targeting dTGD.

**Interference of AP-3 with FDTS activity reduced the building block supply for DNA synthesis**. The flavin-dependent thymidylate synthase (FDTS) executes a multi-step reaction pattern, which catalyzes the reductive methylation of 2′-deoxyuridine-5′-monophosphate (dUMP) to 2′-deoxythymidine-5′-monophosphate (dTMP). During the methylation process, FDTS initially catalyzes the conversion of FAD to $FADH_2$ with the consumption of two-equivalents of NADPH, and then transfers a methylene group from 5,10-methylenetetrahydrofolate (mTHF) to dUMP to generate dTMP and 7,8-dihydrofolate. Once the mTHF is absent in the reactions under $O_2$ existing conditions, FDTS acts as an NADPH oxidase and catalyzes $FADH_2$ and two-equivalents of NADP to produce FAD and $H_2O_2$ (Supplementary Fig. 16)[22].

To characterize the interaction between FDTS and AP-3, recombinant FDTS was obtained by heterologous expression in *E.*

*coli* (Supplementary Fig. 11). Based on SPR biosensor analysis, the dissociation constant $K_D$ of AP-3 with FDTS was determined to be $6.65 \times 10^{-5}$ M, indicating a high affinity of AP-3 for FDTS (Fig. 4a, Supplementary Table 6). To identify the residues critical for AP-3 binding, QG-YNE-labeled FDTS was digested and analyzed by LC-MS/MS. Peptide 90-HFSYSQLSQR-99 was found to be covalently modified and exhibited a mass increase of 712.31 (equivalent to QG-YNE with loss of $N_2$) compared to the unmodified peptide fragment. The amino acid sequence was confirmed unambiguously according to the numerous b-type ions formed after high-energy collisional dissociation (HCD) fragmentation. Further analysis of the MS2 spectrum located the labeling to residue Leu96 (Fig. 4b, Supplementary Fig. 17).

Subsequently, a homo-tetramer structure of the FDTS was simulated with the AlphaFold2 server. The homologous proteins and ThyX from *Thermotoga maritima* (PDB entry 7ndz) displayed high similarity with a RMSD of 1.67 and the highly conserved Ser97 and Tyr99 in active sites (Supplementary Fig. 18). The molecular docking analysis showed that AP-3 bound to a region neighboring the catalytic active pocket of FDTS with an

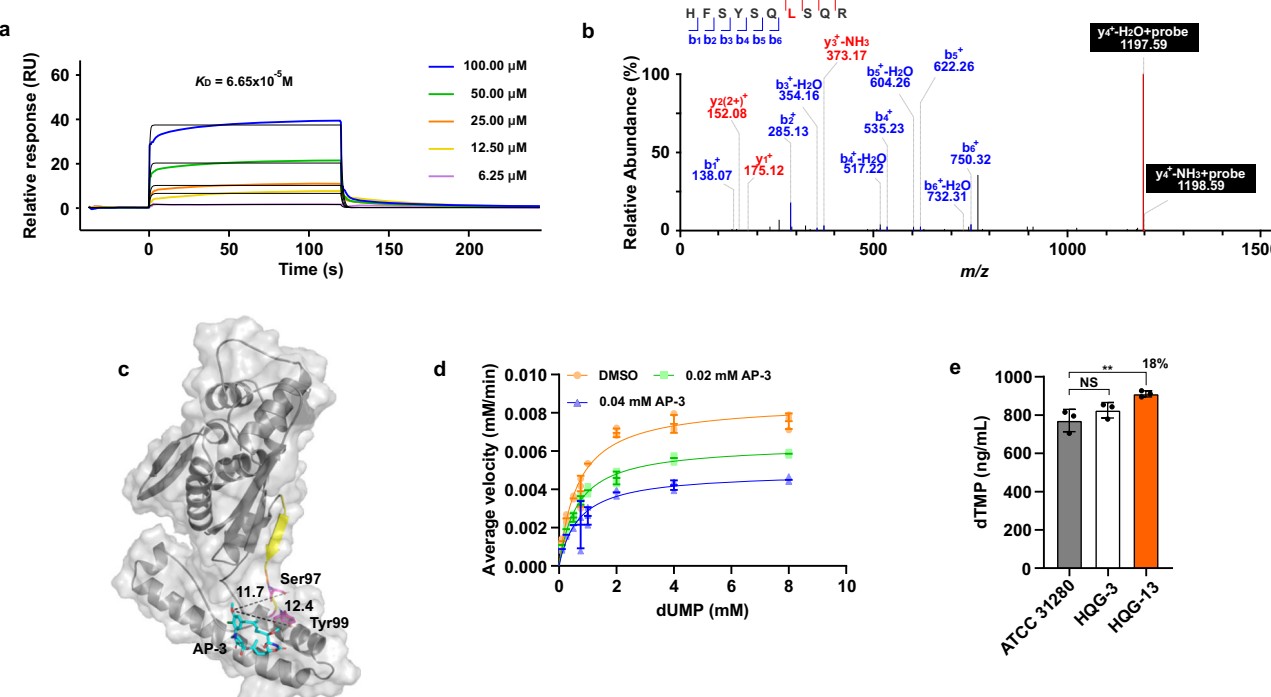

**Fig. 4 AP-3 targets FDTS responsible for the dTMP formation for DNA synthesis. a** Interaction between AP-3 and FDTS was detected by SPR, yielding a $K_D$ value of $6.65 \times 10^{-5}$ M. **b** MS2 spectrum of a QG-YNE-modified peptide from FDTS (90-HFSYSQLSQR-99 (L96 + 712.31 Da)). **c** Overall view of the AP-3 and FDTS homo-tetramer complex. The distances (Å) between the conserved amino acids and AP-3 are indicated by dashed lines. Light grey, FDTS homo-monomer; sky blue, AP-3; yellow, peptide labeled by QG-YNE; orange, amino acid modified by QG-YNE; violet, conserved amino acids of FDTS. **d** Inhibitory kinetics of AP-3 with FDTS are displayed by double reciprocal plots obtained using 0.02 or 0.04 mM AP-3 or 1% (v/v) DMSO (the control). Data are presented as mean values with error bars representing S.D. from three independent experiments. **e** Intracellular dTMP concentrations from strains ATCC 31280, HQG-3 or HQG-13 by quantitative analysis. The discrepancy is shown as percentage numbers. $^{**}P < 0.05$.

affinity of $-6.76$ kcal/mol. AP-3 was anchored in the complex through hydrogen-bond and hydrophobic interactions with key amino residues. The distances between the hydroxyl group at C20 of AP-3 and Ser97 and Tyr99 of FDTS were 11.70 Å and 12.40 Å, respectively (Fig. 4c). The hydroxyl group at C20 was closed to Leu96 with a 3.80 Å distance, facilitating the photoaffinity group connected with the Leu96 during the UV irradiations (Supplementary Fig. 19).

NADPH oxidase activity of FDTS was analyzed with FAD as the H$^+$ acceptor and in the presence of dUMP. Conveniently, NADPH was detectable through linear absorption at 340 nm in a range of 0.015 to 1 mM (Supplementary Fig. 20). The $k_{cat}$ and $K_m$ values of FDTS for the reduction of NADPH were determined to be 57.63 min$^{-1}$ and 0.66 mM, respectively, by monitoring the reactions with different concentrations of dUMP. As the catalytic ability of FDTS was inhibited by AP-3 concentrations above 0.08 mM, reactions were supplemented with AP-3 at final concentrations of 0.02 or 0.04 mM. The $k_{cat}$ and $K_m$ values were further calculated as 42.88 min$^{-1}$ and 0.65 mM in the presence of 0.02 mM AP-3, respectively, and 33.03 min$^{-1}$ and 0.69 mM with 0.04 mM AP-3, respectively (Fig. 4d). These results showed that AP-3 was a non-competitive repressor of FDTS, with a $K_I$ value of 0.05 mM (Supplementary Table 8).

To further explore the physiological contribution of FDTS, we measured the intracellular dTMP concentrations from strains ATCC 31280, HQG-3 and HQG-13. The concentration of dTMP in strain HQG-13 was increased by 14.3% and 8.3% over levels in strains ATCC 31280 and HQG-3, respectively (Fig. 4e, Supplementary Fig. 21). Therefore, AP-3 production impedes the catalytic activity of FDTS and leads to a reduced dTMP supply for DNA synthesis in the producing strain.

**Interaction of AP-3 with ALDH resulted in reduced accumulation of intracellular acetyl-CoA.** ALDHs are commonly involved in the conversion of aldehydes, generated during cellular metabolic processes, into acids with the assistance of cofactor NAD(P)$^+$. Most aldehydes are chemically active and physiologically toxic for the generating organisms, and ALDHs participate in aldehyde detoxification and assist in maintaining the reservoir of reducing equivalents[23,24]. To characterize the interactions between ALDH and AP-3, we purified soluble ALDH from the cell lysates of *Escherichia coli* BL21(DE3) recombinants (Supplementary Fig. 11), and the affinity of AP-3 for ALDH was determined by SPR biosensor analysis. The $K_D$ value was determined to be $1.97 \times 10^{-5}$ M for the interaction of AP-3 with ALDH (Fig. 5a, Supplementary Table 6). To identify the ALDH residues bound by AP-3, recombinant ALDH was photo-labeled with QG-YNE, digested by trypsin and analyzed via LC-MS/MS. The peptide 18-SRYDHFIGGEFTAPAK-33 exhibited a mass increase of 712.31 (equivalent to QG-YNE with loss of N$_2$), and MS2 spectrum analysis of this peptide localized the labeling to Arg19 (Fig. 5b, Supplementary Fig. 22).

Then, a homo-tetramer architecture of the ALDH was constructed through the AlphaFold2 server. Comparison of homologous proteins and ALDH from bovine mitochondria (PDB entry 1a4z) showed a high similarity with a RMSD of 0.82 and conserved Cys302 and Glu263 key residues in the catalytic active sites (Supplementary Fig. 23). Analysis of the molecular docking complex revealed that AP-3 deviated from the catalytic active pocket of ALDH with an affinity of $-7.40$ kcal/mol. The distances between the hydroxyl group at C20 of AP-3 and Cys302 and Glu263 of ALDH were 38.40 Å and 40.50 Å, respectively (Fig. 5c). Hydrogen-bond interactions were formed between the

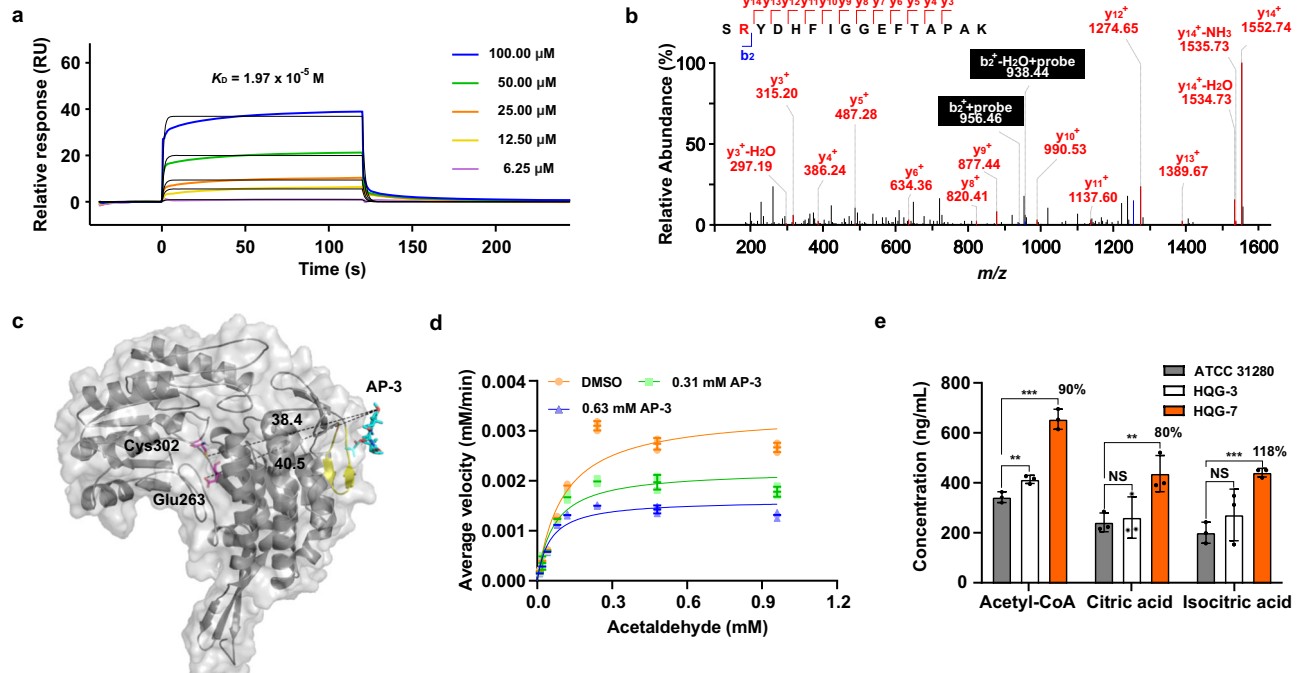

**Fig. 5 Interaction between AP-3 and ALDH involved in central carbon metabolism. a** Interaction between AP-3 and ALDH was detected by SPR, with a $K_D$ value of $1.97 \times 10^{-5}$ M. **b** MS2 spectrum of a QG-YNE-modified peptide from ALDH (18-SRYDHFIGGEFTAPAK-33 (Arg19 + 712.31 Da)). **c** Overall view of the AP-3 and ALDH homo-monomer complex. The distances (Å) between the conserved amino acids and AP-3 are indicated by dashed lines. Light grey, dTGD homo-monomer interacting with AP-3; sky blue, AP-3; yellow, peptide labeled by QG-YNE; orange, amino acid modified by QG-YNE; violet, conserved amino acids of ALDH. **d** Inhibitory kinetics of AP-3 with ALDH are displayed by double reciprocal plots obtained using 0.31 or 0.63 mM AP-3 or 1% (v/v) DMSO (the control). Data are presented as mean values with error bars representing S.D. from three independent experiments. **e** Intracellular acetyl-CoA, citric acid and isocitric acid concentrations from strains ATCC 31280, HQG-3 or HQG-7 by quantitative analysis. The discrepancy is shown as percentage numbers. $^{*}P < 0.1$; $^{**}P < 0.05$; $^{***}P < 0.01$.

carbonyl group of the acetyl moiety at C3 and Arg17 and between the hydroxy group at C9 and Arg17, with the distances of 3.30 Å and 2.80 Å, respectively. The macrolactam ring of AP-3 is attached to the pocked of ALDH through hydrophobic interactions. The hydroxyl group at C20 approached to Arg19 with a 4.20 Å distance, and such distance allowed a connection between the photoaffinity group of QG-YNE probe with Arg19 of ALDH during the UV irradiations (Supplementary Fig. 24).

The enzymatic reactions of ALDH were then established with aldehyde as the substrate and NAD$^+$ as the cofactor. During the transformation of aldehydes into acids by ALDH, the equivalent NADH was generated, which could be conveniently detected using a chromogenic assay (Supplementary Fig. 25). The $k_{cat}$ (8.40 min$^{-1}$) and $K_m$ (0.10 mM) values for the catalytic activity of ALDH on aldehydes were determined, and then AP-3 interference trials with ALDH were carried out. The $k_{cat}$ and $K_m$ values were calculated as 5.48 min$^{-1}$ and 0.07 mM, respectively, with 0.31 mM of AP-3 in the reactions. $k_{cat}$ and $K_m$ values of 3.99 min$^{-1}$ and 0.05 mM were further detected in the presence of 0.63 mM AP-3 (Fig. 5d). These results suggested that AP-3 served as an inhibitor for ALDH, with an inhibition value of $K_I = 0.82$ mM (Supplementary Table 9).

Furthermore, overexpression of the ALDH-encoding gene in strain HQG-7 enhanced the accumulation of intracellular acetyl-CoA by 90% over levels in strain ATCC 31280, and this increased acetyl-CoA consequently enhanced isocitric acid and citric acid accumulation by 118% and 80%, respectively, in HQG-7 in comparison to that of strain ATCC 31280, boosting the growth of HQG-7 with ALDH overexpression. In addition, the intracellular acetyl-CoA, citric acid and isocitric acid levels in the *ansa*-cluster-disrupted strain HQG-3 were also measured and found to be

increased by 34%, 8% and 21%, respectively, in comparison to that of strain ATCC 31280 (Fig. 5e, Supplementary Fig. 26, 27).

**AP-3 production boosted by target engineering**. Since the strains that overexpressed the ALDH, FTDS or dTGD genes displayed markedly increased biomass, when compared to the control strain, during the challenge with high concentrations of AP-3, we further examined the AP-3 production in these recombinants. Strain ATCC 31280, HQG-7, HQG-13 and HQG-17 were cultured in YMS medium, and then AP-3 was extracted from the culture broths. AP-3 titers were determined to be $45.03 \pm 0.95$, $50.33 \pm 5.46$, $71.95 \pm 6.95$ mg/L and $83.47 \pm 4.83$ mg/L for strains ATCC 31280, HQG-7, HQG-13 and HQG-17, respectively, i.e., 11.77%, 59.78% and 85.37% AP-3 titer increases by the respective overexpression of ALDH, FDTS and dTGD (Fig. 6a, Supplementary Fig. 28).

Given the increased AP-3 titers by overexpression of targets, the overexpression constructs *APASM_5765* (FDTS), *APASM_6307* (dTGD) and *APASM_1052* (ALDH) were introduced into an AP-3 high-yielding strain WXR-24 and generated mutants HQG-23, HQG-24 and HQG-25, respectively. After fermentation in YMV medium, AP-3 titers were measured and found to be $225 \pm 33.25$, $395.5 \pm 31.02$, $348.33 \pm 22.49$ and $254 \pm 22.1$ mg/L for WXR-24, HQG-23, HQG-24 and HQG-25, respectively (Fig. 6b).

## Discussion

Microbial bioactive natural products mediate ecological beneficial functions and can confer evolutionary advantages over rival species[25]. For example, thiopeptides act as specific modulators of microbial phenotypes to trigger biofilm formation[26]; at

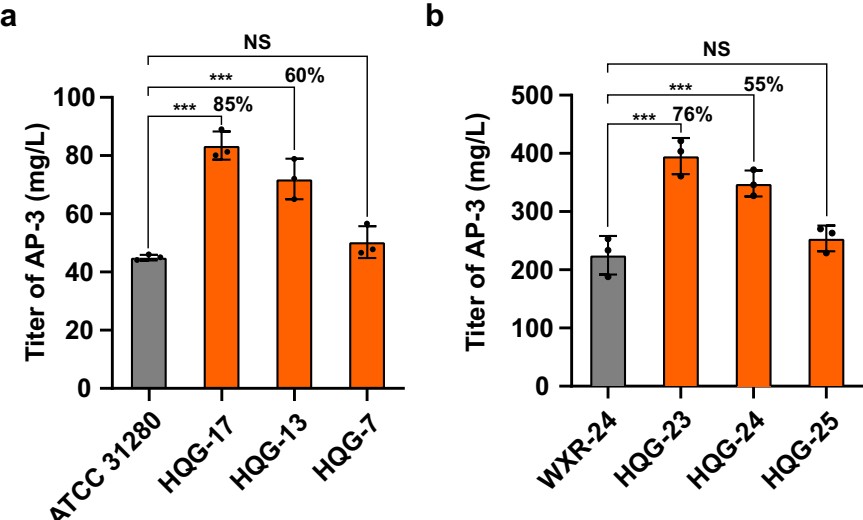

**Fig. 6 AP-3 titer increases by overexpression of target proteins in wild-type and high-yielding strains.** Effect of enhanced expression of target proteins on AP-3 titers in **(a)** wild-type strain ATCC 31280 and **(b)** high-yielding strain WXR-24. The discrepancy is shown as percentage numbers. ***$P < 0.01$.

sub-inhibitory concentrations, tobramycin, tetracycline and norfloxacin function as signaling molecules to monitor the homeostasis of microbial communities and can induce production of bioactive compounds to inhibit protozoan grazing[27]. However, the effects of natural product biosynthesis on their producing strains—be they beneficial or deleterious—seem to be largely unknown. In our work, we confirmed the biosynthesis of ansamitocins incurred stress with reduced biomass to the producing strain, as shown either by supplementation of exogenous AP-3 at high concentrations to the cultures[14] or by deleting the ansamitocin gene cluster in the wild-type (Fig. 2b). Furthermore, we did systematic investigation by structure simulation and binding protein capture with AP-3 derived photoaffinity probe, and identified FtsZ, the cytokinetic Z-ring protein essential for cell division[14], and deoxythymidine diphosphate glucose-4,6 dehydratase (dTGD), flavin-dependent thymidylate synthase (FTDS) and aldehyde dehydrogenase (ALDH) as the indigenous targets of AP-3, which are largely responsible for the stress caused by AP-3 biosynthesis. Similar stress effects were observed in opportunistic pathogen *Pseudomonas aeruginosa*[28] and marine bacterium *Phaeobacter inhibens*[29]. The pyocyanin-producing wild-type *P. aeruginosa* PA14 had a higher cell death ratio when compared to the Δ*phz* mutant with pyocyanin biosynthetic gene cluster deleted, and exogenous supplementation of pyocyanin increased the cell death ratio of the Δ*phz* mutant[28]. In *Phaeobacter inhibens*, inactivation of the tropodithietic acid biosynthetic genes led to a higher biomass and a higher carbon usage efficiency of the mutant than the wild-type[29]. Subsequent transcriptomic and proteomic comparisons revealed upregulated expressions of branched-chain amino transferase IlvE, several uptake systems for amino acids and inorganic nutrients, and some components of the respiratory chain[30].

Chemoproteomic approaches have been successfully used to identify the binding proteins of microbiota primary metabolites (fatty acids, aromatic amino acid-derived metabolites, vitamins, bile acids etc.)[31], mycolic acids in mycobacteria[32], bile acid in *Clostridioides difficile*[33], and heme in Gram-positive and Gram-negative bacteria[34]. To our knowledge, however, chemoproteomics has not previously been deployed to explore the binding spectrum of bioactive compounds in the producers. Whereas the activity-based protein profiling (ABPP) methodology uses chemical probes to specifically label catalytically active proteins[35], the photoaffinity approach explores probes with photoreactive

moieties, such as the diazrine group, to interact and form crosslinks with any specific binding proteins[36]. As shown in our experiments, three crosslinking reactions for the QG-YNE probe and cellular proteins resulted in efficient and specific labeling with 14 shared captured proteins (Fig. 2c), among which Asm7 is with catalytic activity toward chloro-proansamitocin[16], and AP-3 probably exerts allosteric effects on dTGD, FTDS, and ALDH (Figs. 3d, 4d, 5d). Therefore, chemoproteomics is indeed a powerful strategy for characterizing the proteome-wide interactions of bioactive compounds in their producers.

We have developed many genetic strategies for AP-3 titer increase, including alleviation of rate-limiting post-PKS biosynthetic steps[37], attenuation of the mycelial fragmentation[15], improved AP-3 exportation through overexpression of export genes[38], and strengthened resistance by overexpression of the target protein FtsZ[14] with titer increases between 24.94% and 93% and up to 330.6 mg/L. In strains that overexpressed the AP-3 target proteins, we found that titers of AP-3 were boosted up to 85% and 76% over levels in wild-type strain ATCC 31280 and high-yielding strain WXR-24, respectively (Fig. 6). This result indicated that the expression of target proteins in antibiotic producing strains is necessary for a designated antibiotic, and should be considered for titer improvement.

Interestingly, while dTGD and FTDS overexpression substantially enhanced the production of AP-3, the overexpression of ALDH only led to promoted growth of the producing strain and had no effect on AP-3 production. ALDH oxidizes acetaldehyde to acetic acid, which can be further transformed into acetyl-CoA. Acetyl-CoA is not only a key molecule in the central carbon metabolism of microbes[39], but also enzymatically converted to the malonyl-CoA extender units for AP-3 biosynthesis[40]. Notably, the intracellular concentration of acetyl-CoA was 90% higher in HQG-7 overexpressing ALDH, than that in strain ATCC 31280 (Fig. 5e). To explore the utilization of acetyl-CoA, we further examined key intermediates in the TCA cycle and found that intracellular concentrations of isocitric acid and citric acid were dramatically enhanced by 118% and 80%, respectively, in HQG-7 when compared with strain ATCC 31280. Therefore, the increased acetyl-CoA flux in the ALDH overexpression mutant is most likely diverted to the TCA cycle and biomass accumulation, rather than the biosynthesis of AP-3.

In summary, our current research concentrated on the mining of cryptic targets of the bioactive secondary metabolite AP-3 in its

naturally producing strain through the power of chemoproteomics and an MS-based quantitative approach. AP-3 was found to specifically bind with three protein targets and functioned as their non-competitive inhibitor. Our findings further suggested that AP-3 production causes a physiological stress on the producer, which could be partly relieved by overexpression of the target proteins. Additionally, enhancing the expression of these targets also achieved titer improvement of AP-3. This integration of target screening and engineering exhibits great potential for illuminating the biological effects of bioactive secondary metabolites and their cryptic targets, and for improving strains for industrial application, including the production of antibiotics.

## Methods

**Strains, plasmids and media**. Detailed information on culture media, strains, plasmids and primers is provided in Supplementary Tables 1–4. *Escherichia coli* strains DH10B, ET12567(pUZ8002) and BL21(DE3) were used as hosts for cloning, *E. coli*-*Actinosynnema* bi-parental conjugation and protein expression, respectively. Plasmid pJTU1278 was used to construct the gene disruption mutant. The integrative vector pLQ648, containing the constitutive strong promoter *kasO*p*, was adapted to overexpress target protein-encoding genes, and pET30a was used for protein expression.

During fermentation, strain ATCC 31280 and recombinants were firstly cultured on YMG plates for 48 h at 30 °C, then the mycelia were transferred into S1 medium at 30 °C with shaking at 220 r.p.m. and cultured for 24 h. Subsequently, S1 culture was inoculated into S2 medium at a final concentration of 3.3% (v/v) and cultivated at 30 °C and 220 r.p.m. for another 24 h. Finally, S2 seed broth was transferred into YMS medium at a final concentration of 10% (v/v) and cultivated at 25 °C and 220 r.p.m. for 7 days to detect the biomass and AP-3 titer estimation in YMS/YMV media.

**Quantitative analysis of AP-3 production by liquid chromatography-mass spectrometry (LC-MS)**. For quantitative analysis of AP-3 (Supplementary Fig. 28) titers, an equal volume of methanol was supplemented into the final fermentation cultures and stirred. The mixture was sonicated for 30 min. Then, 1 mL of the mixture was collected and centrifuged, and the supernatant was filtered through a 0.22 μm filter and subjected to LC-MS analysis. For LC-MS determination, samples were analyzed on an Agilent series 1290-MS 6230 system (Agilent Technologies, Santa Clara, CA, USA) equipped with an Agilent Eclipse Plus C18 column (4.6 × 250 mm, 5 μm), with elution using 45% A (H$_2$O with 0.5% formic acid): 55% B (acetonitrile) for 35 min and detection at 254 nm. The ion source was an AJS ESI model with positive polarity. The gas temperature, flow and nebulizer pressure were 325 °C, 8 L/min and 35 psi, respectively. The fragmentor, skimmer and OCT 1 RF Vpp were set at 175, 65 and 750 V, respectively. All derivatives of AP-3 were detected or isolated by this method. The *m/z* of related compounds were calculated by high-resolution mass spectra and listed in corresponding legends of supplementary figure.

**Synthesis of an AP-3-derived photoaffinity probe**. 20-Demethyl-ansamitocin P-3 (QG-1) (46.5 mg, 0.075 mmol) was dissolved in anhydrous dimethyl formamide (DMF) (1.0 mL), and then K$_2$CO$_3$ (20 mg, 0.15 mmol) and diazirine-alkyne moiety (YNE[41]) (synthesized by Bioduro-Sundia, Shanghai, China, 3.1 mg, 0.0125 mmol) were successively added into the solution. After 1 h of stirring at 50 °C, the reaction was detected by thin-layer chromatography with mobile phase using 95% dichloromethane and 5% methanol, and additional YNE (4.4 mg,

0.0178 mmol) was added to the reaction mixture followed by incubation at 50 °C for another 3 h in the dark. The mixture was washed with saturated Na$_2$CO$_3$ and further dealt with 1 mol HCl and brine. The organic layer was dried by anhydrous Na$_2$SO$_4$ and then filtered and concentrated. The crude extract was purified by high-performance liquid chromatography (HPLC) to obtain the desired product, QG-YNE (18 mg, 80.6% yield). $^1$H and $^{13}$C nuclear magnetic resonance (NMR) spectra were recorded with a Bruker Advance 400 MHz in CDCl$_3$ ($^1$H NMR, 400 MHz; $^{13}$C NMR, 100 MHz) spectrometer using tetramethylsilane as internal standard.

**Mass spectrometry-based affinity-based protein profiling**. The procedure for target protein capture consisted of harvesting the mycelia of strain ATCC 31280 on the third day by centrifugation at 3488 × *g* (Thermo, TX-750), followed by washing with 7% NaCl, suspending in PBS buffer (150 mM NaCl, 3 mM KCl, 8 mM Na$_2$HPO$_4$, 2 mM KH$_2$PO$_4$, pH 7.2–7.4) and crushing with an ultrasonic processor. The 10 mM chemical probe (QG-YNE) was pre-incubated with 50 μL of 30 mg/mL cell lysate for 2 h and then irradiated with UV light at 365 nm for another 20/ 10/ 5 mins on ice for the generation of the covalent binding between the diazirine group on the probe and the amino acid residues of the targeted proteins. The probe-binding proteins underwent another click reaction with 0.5 mM biotin-N$_3$ assisted by 0.1 mM Tris-hydroxypropyltriazolylmethylamine (THTPA), 1 mM CuSO$_4$ and 1 mM vitamin C sodium (NaVc). Then, 50 μL streptavidin-sepharose (GE Healthcare) beads were poured into each reaction sample and incubated for 16 h with continuous rotation at room temperature. The beads were washed with 600 μL 0.1 M PBS, 0.5 M NaCl, 4.0 M urea, and 100 mM triethylamonium bicarbonate (TEAB) consecutively. The enriched proteins were subjected to reductive alkylation with 200 μL 10 mM Tris(2-carboxyethyl)phosphine (TCEP) and 200 μL 55 mM iodoacetamide. Enriched proteins were digested with 0.25 μg trypsin (Promega) overnight at 37 °C. The digests of both dead-YNE (diazirine-alkyne moiety, Supplementary Fig. 6) and QG-YNE treated-samples were labeled with TMT$^2$−126 and TMT$^2$−127 Isobaric Label Reagent (Thermo Scientific), respectively. The labeled sample was added to a Thermo Acclaim PepMap RSLC analytical column (75 μM × 15 cm) and washed by 0.1% formic acid in H$_2$O and acetonitrile in 0.3 μL/min. Then, the peptide fragments were analyzed on a Thermo Orbitrap Fusion Lumos proteomic mass spectrometer (Thermo Scientific)[42].

**Gene knockout and overexpression**. The *asm7* knockout mutant HQG-1 was obtained from strain ATCC 31280 using a homologous recombination approach[43]. The left and right homologous arms were amplified by PCR using primer pairs deasm7-L-F/R and deasm7-R-F/R, respectively. After sequencing validation, the DNA fragments were cloned into a pJTU1278-derived shuttle vector at the *Bam*HI/*Eco*RI and *Eco*RI/*Hin*dIII sites using a DNA Ligation Kit (TaKaRa). The constructed plasmids were sequenced, introduced first into *E. coli* ET12567(pUZ8002) by transformation and then into strain ATCC 31280 by conjugation. After culturing as a lawn of YMG with 0.2 mg/mL nalidixic acid and thiostrepton for 7 days at 30 °C, the exconjugants were further cultured without thiostrepton for two rounds on YMG solid medium. Then, the double crossover mutants were screened by PCR to identify the target recombinants. The *asm7* deletion mutant was named HQG-1, and the *asmA* disruption recombinant HQG-3 was obtained in the same way as HQG-1.

Genes coding for the target proteins were overexpressed using a pSET152-derived integrative vector containing the strong constitutive promoter *kasO*p*. These genes were amplified by

PCR using corresponding primer pairs (Supplementary Table 3), and after sequencing validation, the target genes were individually inserted into the pSET152-derived shuttle vector at the NdeI/EcoRI sites using the DNA Ligation Kit. The constructed plasmids were first introduced into E. coli ET12567(pUZ8002) by transformation and then into the producing strains ATCC 31280 or WXR-24 by conjugation. After culturing on the lawn of YMG with 0.2 mg/mL nalidixic acid and apramycin for 4 days at 30 °C, the exconjugants were transferred onto YMG plates containing apramycin and nalidixic acid, cultured for 2 days and then validated by PCR.

**Evaluation of the AP-3 tolerance of ATCC 31280 and recombinant strains**. To test AP-3 tolerance, the strains overexpressing the target genes (Supplementary Table 1) and the control strain ATCC 31280 were cultivated on YMG solid media for 2 days. Following the successive cultivation of the seed broth in S1 and S2 media, 0.15 mL of S2 mycelium cultures (diluted to similar numbers of colony-forming units by $OD_{600}$ detection) were transferred into 24-deep-well plates containing a total volume of 1.5 mL YMS medium with a final concentration of 200 mg/L AP-3. Comparison of the biomass of the recombinants and strain ATCC 31280 was carried out after 5 days of fermentation. Evaluation of the tolerance of the related strains against AP-3 was determined by dry cell weight on day 5. Additionally, HQG-7, HQG-13, HQG-17 and strain ATCC 31280 were fermented with a final concentration of 400 mg/L AP-3 in the fermentation shake-flasks to investigate AP-3 tolerance.

**Protein expression, purification and enzyme inhibition assays**. Gene APASM_1052, APASM_3207, APASM_5765 and APASM_6307 coding for ALDH, Asm7, FDTS and dTGD, respectively, in strain ATCC 31280 were amplified by PCR using primer pairs ALDH-F/R, Asm7-F/R, FDTS-F/R and dTGD-F/R (Supplementary Table 3), respectively. The obtained PCR products were inserted into the EcoRI/HindIII sites of the vector pET30a for sequencing validation, and the verified plasmids were introduced into E. coli BL21 for protein expression. The heterologous expression strains were incubated at 37 °C and 220 r.p.m. to an $OD_{600}$ of 0.8 and induced with 0.1 mM isopropyl β–D-1-thiogalactopyranoside (IPTG), then successively cultured for another 10 h at 16 °C. The cells were harvested by centrifugation at $3488 \times g$ (Thermo, TX-750), 4 °C for 15 min, and then the pellet was resuspended in the corresponding buffers: Tris-HCl buffer (300 mM NaCl, 25 Mm Tris, pH 8.0) for dTGD; HEPES buffer (pH 7.5) for FDTS; and Tris-HCl buffer (pH 8.5) for ALDH. After sonication on ice for 10 min with 2 s at 3 s intervals, the cell debris was removed by centrifugation at $13,800 \times g$ (Thermo Scientific IEC MicroCL 21/21 R). and maintained at 4 °C for 30 min. Then the supernatant was applied to a nickel-NTA affinity chromatography column and eluted with resuspension buffer containing a gradient of 50–250 mM imidazole. The 250 mM imidazole eluent was ultra-concentrated to 500 μL, and then resuspension buffer was added to wash away the imidazole, followed by ultra-concentration again to 500 μL. The purified proteins were verified by SDS-PAGE (Supplementary Fig. 11) and used to analyze protein-AP-3 interactions and enzymic catalytic reactions. Enzyme concentration was determined spectrophotometrically at 280 nm using NanoDrop One (Thermo Scientific).

The biochemical reactions of dTGD were performed at 30 °C for 60 min in 100 μL of 50 mM Tris-HCl buffer (pH 8.0) containing 0.5 μg dTGD and 0.5 mM $NAD^+$ with different concentrations of dTDP-D-glucose as the substrate (0.1, 0.16, 0.2, 0.32, 0.48, 0.64 or 0.8 mM). The reaction was terminated by addition of 10 μL of 1 M NaOH. The final product, dTDP-6-deoxy-D-xylo−4-hexulose, was detected using a microplate reader at a wavelength of 320 nm to monitor the reaction[44]. The corresponding $K_m$ and $V_{max}$ values were calculated by measuring the initial reaction rates of dTGD under different concentrations of dTDP-D-glucose. Based on this analysis platform, 0.31 and 0.63 mM AP-3 were supplemented into the reactions to calculate the inhibition constants ($K_I$) of AP-3 for dTGD.

The catalysis reactions of ALDH were carried out at 30 °C for 60 min in 100 μL of 50 mM Tris-HCl buffer (pH 8.5) containing 2.5 μg ALDH, 100 mM KCl, and 2 mM $NAD^+$ with 0.01, 0.02, 0.04, 0.08, 0.12, 0.24, 0.48 or 0.96 mM acetaldehyde as the substrate. Then, 0.1 mM water-soluble tetrazole salt (WTS-8) and 1 μM 1-methoxy-5-methylphenazinium methyl sulfate (1-mPMS) were added to the mixture to complete the reactions. With the electron trapping agent of 1-mPMS, WTS-8 can react with NADH to generate formazan. The concentration of formazan was quantified by a microplate reader at a wavelength of 450 nm ($A_{450}$)[45] (Supplementary Fig. 25). The reaction constants of $K_m$ and $V_{max}$ were assessed by the absorption of $A_{450}$ catalyzed by ALDH for different concentrations of acetaldehyde. Additionally, 0.31 and 0.63 mM of AP-3 were added to the reactions to calculate the inhibition constants ($K_I$) of AP-3 for ALDH.

Reactions with FDTS were conducted at 30 °C for 25 min in 100 μL of 50 mM HEPES buffer (pH 7.5) containing 0.5 μg FDTS, 1 mM $MgCl_2$, 62.5 μM FAD, and 0.6 mM NADPH with 0.1, 0.25, 0.5, 0.75, 1, 2, 4 or 8 μM dUMP as the substrate. The reactions were stopped by addition of 10 μL of 600 μM formic acid. The reaction constants of $K_m$ and $V_{max}$ for FDTS were determined by the consumption of NADPH to NADP using $A_{340nm}$ detection[46]. Additionally, 0.02 and 0.04 mM of AP-3 were added into the reactions to calculate the inhibition constants ($K_I$) of AP-3 for FDTS.

**Surface plasmon resonance biosensor analysis**. Surface plasmon resonance biosensor analysis system was performed with a Biacore TM 8 K (GE Healthcare) for monitoring the biomolecular interactions at 25 °C. The buffer containing the purified proteins was exchanged with PBS for preventing interference to the detection signal during the test. Then, the proteins were diluted to a concentration of 10 μg/mL with 10 mM sodium acetate buffer (pH 4.0) and then immobilized on the surface of CM5 sensor chips. AP-3 was dissolved in PBS buffer with 0.5% DMSO and diluted to gradient concentrations of 100, 50, 25, 12.5 and 6.25 μM. During the binding process, AP-3 solution flowed across the CM5 surface in PBS buffer at a flow rate of 30 μL · min⁻¹ for 90 s. Then in the dissociation step, PBS buffer was passed across the surface of CM5 at a flow rate of 30 μL · min⁻¹ for 120 s to elute AP-3. For every purified protein, responses signals from experiments with gradient concentrations of AP-3 were recorded, and the affinity constant ($K_D$) was calculated by Biacore Insight Evaluation Software.

**Microscopy observation**. Cell morphology was characterized by scanning electron microscopy (HITACHI S340II). The harvested mycelia were washed three times with 0.1 M PBS buffer and fixed in 2.5% glutaraldehyde (dissolved in PBS buffer) for 12 h. Fixed cells were dehydrated with gradient concentrations of 5%, 15%, 30%, 60%, 80%, 90%, 95% and 100% ethanol solution (mixed with PBS buffer), and ethanol was removed using an automatic critical point dryer (LEICA). Dried mycelia were fixed on the copper platform with conductive adhesive for surface gold-plating in a vacuum coater (LEICA). The images were captured with

10 k-fold magnification under scanning electron microscopy. The specimen size and height were set as 51 mm and 1 mm.

**Identification of QG-YNE-binding peptides**. Ten micrograms of recombinant His-tagged proteins ALDH, FDTS and dTGD in 100 μL PBS buffer were incubated with 10 μM of QG-YNE at room temperature for 2 h, followed by 365 nm UV irradiation for 20 min on ice. QG-YNE-labeled proteins were separated by SDS-PAGE, further extracted and dissolved in 50 mM NH4HCO3 buffer. Chemical modification by reducing with 20 mM dithiothreitol at 60 °C for 60 min and alkylating with 50 mM iodoacetamide at 30 °C for 30 min were performed to reduce the disulfide bonds and generate an open conformation of the proteins[47]. Then, the samples were digested with 2 μg trypsin (Thermo Scientific) at 37 °C for 12 h to generate peptide fragments. The peptide fragment samples were desalted with Ziptip desalting columns (Pierce) and resuspended in 10 μL ddH2O containing 0.1% formic acid after drying by evaporation. The final samples were analyzed using a Thermo Easy nLC1200/Q Exactive plus proteomic mass spectrometer to identify the amino acid residue(s) bound by the photoaffinity probe.

**Extraction and detection of intracellular metabolites**. After three days of fermentation, mycelia were harvested, and the intracellular metabolites were quantitatively analyzed. Specifically, 25 mL of fermentation broth was collected and centrifuged at $6200 \times g$ (Thermo, F15-6×100y). for 12 min. The mycelia were washed with 25 mL of PBS buffer three times. Then, the washed mycelia were resuspended in 25 mL of distilled water and divided equally into 5 tubes, followed by centrifugation at $13,800 \times g$ (Thermo Scientific IEC MicroCL 21/21 R). for 5 min and removal of the supernatant. Then, 2.5 mL of extracting solution containing 45% acetonitrile, 45% methanol and 10% glacial acetic acid was added, and the cell pellet was resuspended and sonicated for 15 min. After two extractions, the combined mixture was centrifuged at $13,800 \times g$ (Thermo Scientific IEC MicroCL 21/21 R). for 5 min, and the supernatant was filtered through a 0.22 μm filter and collected for analysis of the intracellular metabolites. All steps above were performed at 4 °C or on ice.

The quantitative analysis of acetyl-CoA was performed using an LC-MS 1260-QQQ 6470 (Agilent) with an Eclipse-C18 (4.6 × 250 mm, 5 μm, Agilent) column at a flow rate of 0.5 mL/min with a 2 μL injection volume. The mobile phase was composed of (A) 20 mM ammonium acetate in Milli-Q water with a pH of 7.4 and (B) methanol. The metabolites were eluted by gradient mobile phase as follows: 0–5 min, 2% B; 5–15 min, 2% B to 50% B; 15–20 min, 90% B; 20–25 min, 90% B to 2% B; and 25–30 min, 2% B. Quantitative analysis of citric acid, isocitric acid and dTMP was conducted using an ACQUITY UPLC H-Class system & Xevo TQ-XS triple quadrupole mass spectrometer (Waters, Milford, MA, USA) with ACE Excel C18-PFP (100 × 2.1 mm, 1.7 μm, Phenomenex) at a flow rate of 0.3 mL/min with 1 μL injection volume. Water and acetonitrile, both containing 0.1% formic acid, were used as mobile phases A and B, respectively, with a linear gradient of 1–100% B within 10 min.

Metabolites were scanned by multiple reaction monitoring (MRM) mode with a rate of 0.01 s/scan and a capillary voltage set at 1 kV. The source temperature was held at 150 °C while that of the desolvation gas was 450 °C. The flow rates of the desolvation gas (nitrogen) and cone gas (argon) were 900 and 50 L/h, respectively. Acetyl coenzyme A was detected in positive ion and MRM modes. dTMP, citric acid and isocitric acid were detected in negative ion and MRM modes. The calibration curves and MRM quantitative signal for quantification of dTMP are shown in Supplementary Fig. 21; for acetyl-CoA, in Supplementary Fig. 26; for citric acid and isocitric acid, in Supplementary Fig. 27. The cone voltage (V) and collision energy (eV) of acetyl-CoA, citric acid, isocitric acid and dTMP were set as 20/15, 20/15, 20/20 and 15/20, respectively.

**Evaluation of binding models for AP-3 and target proteins**. Protein modeling was performed by AlphaFold v2.1.1[48] and ranked_0.pdb, which contained the prediction with the highest confidence and selected for further analysis. All proteins were modeled in both the monomer format and homo-multimer format. To determine oligo states, amino sequences of ALDH, FDTS and dTGD were used to search the PDB database (https://www.rcsb.org/), and homologous proteins were analyzed. ALDH was aligned to 5gtl with a sequence identity of 43%, which forms a homo-tetramer. FDTS and dTGD were aligned to 3hzg and 1r66, which form a homo-tetramer and homo-dimer, respectively, with sequence identities of 72% and 70%, respectively. Protein structures were analyzed by PyMol (Version 2.3.0)[49].

The molecular structure of AP-3 was obtained from the PDB database (PDB ID: 7e4p). AutoDock Tools[50] was employed to convert protein and molecular files into pdbqt with added polar hydrogen atoms. Molecular docking was performed by AutoDock Vina 1.2.3[51] with AP-3 and target proteins as the ligand and receptors, respectively. At the same time, the argument exhaustiveness was set to 32 to give a more reliable docking result. The binding sites of AP-3 were visualized by PyMol (Version 2.3.0). The hydrogen bond interactions and hydrophobic interactions between AP-3 and proteins were analyzed by LigPlot+ (Version v2.2.5)[52] (Supplementary Figs. 14, 19 and 24).

**Statistics and reproducibility**. One-way ANOVA test was performed for the statistical analysis of biomass, mycelium length, AP-3 titer and concentrations of intracellular metabolites. $P \geq 0.1$, $P < 0.1$, $P < 0.05$, and $P < 0.01$ were defined as insignificant, significant, moderately significant, and highly significant, respectively. CONFIDENCE.T formula in Excel was used to estimate the 95% confidence interval of mean. Cohen's $d$ was employed to calculate the effect size of mean. Three independent biological replicates were used for the experiments of biomass measurement and metabolites quantification, while five samples were chosen for the mycelium length measurement. The effects of AP-3 and probe QG-YNE on the growth rate of yeast indicator strain were measured for one time. The interactions between AP-3 and deoxythymidine diphosphate glucose-4,6 dehydratase (dTGD), flavin-dependent thymidylate synthase (FDTS) and aldehyde dehydrogenase (ALDH) were measured for one time. Three independent biological replicates were used in Michaelis-Menten constant measurement of three target proteins.

**Reporting summary**. Further information on research design is available in the Nature Portfolio Reporting Summary linked to this article.

# Data availability

The main data supporting the findings of this study are available within the article and its Supplementary Information files. The datasets generated and analyzed during this paper are available from the corresponding authors upon request. The source data underlying the graphs and proteomic charts are uploaded as a Supplementary Data 1 and 2. The sequences for all genes described in this manuscript are available in the GenBank databases under the accession numbers CP073249.1. The mass spectrometry proteomics data have been deposited to the ProteomeXchange Consortium via the PRIDE partner repository with the dataset identifier PXD043867.

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

## Acknowledgements

We dedicate this paper to the memory of our late colleague, Prof. Youli Xiao. We acknowledge the early efforts by Youli Xiao to synthesize the photoaffinity probe and conduct the chemoproteomic analysis. We appreciate the helpful comments from Prof. Shenying Li of Shandong University and Prof. Ningyi Zhou from SJTU. We thank the Core Facility and Technical Service Center for SLSB and the Instrumental Analysis Center in SJTU for data collection. This work was supported by the National Key Research and Development Program of China (grant Nos. 2021YFC2100600 and 2019YFA0905400), National Natural Science Foundation of China (grant No. 31830104), and Science and Technology Commission of Shanghai Municipality (grant Nos.19JC1413000 and 19430750600) to L.B.

## Author contributions

L.Q.B. and Q.G.H. conceived the project. L.Q.B., Q.J.K., and Q.G.H. designed the experiments. Q.G.H., X.Z., X.N.F. and Z.Y.G. performed the experiments. Q.G.H., Q.J.K., Y.L.Z. and L.Q.B. analyzed the data. Q.G.H., Q.J.K., Y.L.Z., and L.Q.B. wrote the manuscript.

## Competing interests

The authors declare no competing interests.
