## [Peer Review File · Communications Biology]

Reviewers' comments:

Reviewer #1 (Remarks to the Author):

The manuscript entitled "Biosynthesis of ansamitocin P-3 incurs stress on the producing strain at multiple targets" found that AP-3 functions as a non-competitive inhibitor of three target proteins that exert physiological stress on the producing strain by interfering with different metabolic pathways, and that overexpression of these target proteins increases the biomass of the strain and significantly increases the production of AP-3. This finding suggests that the identification and engineering of implicit targets for bioactive natural products can lead to a deeper understanding of microbial physiology and improved product yields.

This manuscript requires minor corrections. I have several comments as below.

(1) The scale of the SEM in Figure 1 should be clearly marked, whether one frame represents 5 μm ?

(2) Many of the figures in the article involve statistical data analysis. Therefore, the methods of statistical analysis should be written in detail in the material methods.

(3) Mass spectrometric characterization of AP-3 involved in Figure S2 of the Supplementary. The mass spectra used should be high-resolution mass spectra, please add a relevant note on the material methods.

Reviewer #2 (Remarks to the Author):

Huang and coworkers have authored a study concerning the biological targets for ansamitocin P-3 in the producing organism *Actinosynnema pretiosum* ATCC 31280. The authors' premise that the physiological effects of overproduction of natural products on the producing organisms is underexplored is significant. The authors employed a chemoproteomic approach to identify binding partners of AP-3 within the proteome of *A. pretiosum* ATCC 31280. The authors also generated a photoaffinity probe, QG-YNE, to further probe the protein binding targets via mass spectrometry.

The data that AP-3 binds to deoxythymidine diphosphate glucose 4,6-dehydratase, aldehyde dehydrogenase, and flavin-dependent thymidylate synthase is compelling and comprehensive. The authors also demonstrated convincingly that the inhibition of these proteins leads to physiological stress, and that this stress could be further remedied by overexpressing the target proteins. This was evidenced by increased production of AP-3, and importantly, an increase in biomass.

The supplementary information strongly supports the chemical structural assignment of AP-3, and it provides comprehensive information for the replication of methods and findings. Overall, the work is highly novel and significant, and it illuminates the impact of overproduction of natural products on the growth of the producing organism and provides a mechanistic explanation for this growth inhibition. This study will be of interest to readers in the natural products community and metabolic engineering community, as they work towards the rational engineering of microorganisms for improved production titers of cytotoxic metabolites.

- Minor Revisions

Recommend re-reading the manuscript and correcting minor grammatical and syntax mistakes throughout the manuscript.

Reviewer #3 (Remarks to the Author):

Overall this is a scientifically sound manuscript that is thoroughly reported and addresses a question of what do natural products do via the creation of probes to look at downstream physiological stressors with the annasmitocin pathway from *Actinosynnema pretiosum* ATCC 31280. My comments are generally minor and I think the paper should be published with minor revisions, but I believe the manuscript would be improved with the following additions:

P9 lines 170-171--It might be helpful to comment briefly on which types of metabolic processes that are essential are disrupted (I know examples are later, but it would improve context for the reader).

Figure 2--It says the bar is black in the figure legend, but it is actually grey in panels d and e.

Overall, the text is really small on all the figures, especially Figure 2 which reduces the clarity of the text

Dear reviewers:

We gratefully thank all of you for taking your time and effort to review this manuscript entitled “Biosynthesis of ansamitocin P-3 incurs stress on the producing strain at multiple targets” (ID: COMMSBIO-23-1416-T). All the reviewers’ comments and suggestions are professional and valuable, which have enabled us to improve the quality of our manuscript significantly. We have carefully studied the comments and accordingly revised the manuscript.

Below are the point-by-point responses, and changes made in the revised version are indicated in **RED**.

Responses to the reviewers’ comments:

Reviewer 1

The manuscript entitled “Biosynthesis of ansamitocin P-3 incurs stress on the producing strain at multiple targets” found that AP-3 functions as a non-competitive inhibitor of three target proteins that exert physiological stress on the producing strain by interfering with different metabolic pathways, and that overexpression of these target proteins increases the biomass of the strain and significantly increases the production of AP-3. This finding suggests that the identification and engineering of implicit targets for bioactive natural products can lead to a deeper understanding of microbial physiology and improved product yields.

This manuscript requires minor corrections. I have several comments as below.

1) The scale of the SEM in Figure 1 should be clearly marked, whether one frame represents 5 μ m.

Response: The scale of the SEM image involved in Figure 1c, d and Figure 3e are divided in ten frames with a total length of 5 μ m, and the annotations have supplied in the figure notes in Figure 1 (Lines 128-129, Page 7) and Figure 3 (Lines 286-287, Page 16).

2) Many of the figures in the article involve statistical data analysis. Therefore, the methods of statistical analysis should be written in detail in the material methods.

Response: The detailed statistical analysis is amended as followed in the Methods as suggested (Lines 935-939, Page 46): Significance analysis One-way ANOVA test was performed for the statistical analysis of biomass, mycelium length, AP-3 titer and concentration of intracellular metabolites. $P \geq 0.1$, $P < 0.1$, $P < 0.05$, and $P < 0.01$ were defined as insignificant, significant, moderately significant, and highly significant, respectively.

3) Mass spectrometric characterization of AP-3 involved in Figure S2 of the Supplementary. The mass spectra used should be high-resolution mass spectra, please add a relevant note on the material methods.

Response: As suggested, we have added the following note to the experimental methods: The m/z of related compounds were calculated by high-resolution mass spectra and listed in corresponding legends of supplementary figure (Line 693-695, Page 35).

Reviewer 2

Huang and coworkers have authored a study concerning the biological targets for ansamitocin P-3 in the producing organism *Actinosynnema pretiosum* ATCC 31280. The authors' premise that the physiological effects of overproduction of natural products on the producing organisms is underexplored is significant. The authors employed a chemoproteomic approach to identify binding partners of AP-3 within the proteome of *A. pretiosum* ATCC 31280. The authors also generated a photoaffinity probe, QG-YNE, to further probe the protein binding targets via mass spectrometry.

The data that AP-3 binds to deoxythymidine diphosphate glucose 4,6-dehydratase, aldehyde dehydrogenase, and flavin-dependent thymidylate synthase is compelling and comprehensive. The authors also demonstrated convincingly that the inhibition of these proteins leads to physiological stress, and that this stress could be further remedied by overexpressing the target proteins. This was evidenced by increased production of AP-3, and importantly, an increase in biomass.

The supplementary information strongly supports the chemical structural

assignment of AP-3, and it provides comprehensive information for the replication of methods and findings. Overall, the work is highly novel and significant, and it illuminates the impact of overproduction of natural products on the growth of the producing organism and provides a mechanistic explanation for this growth inhibition. This study will be of interest to readers in the natural products community and metabolic engineering community, as they work towards the rational engineering of microorganisms for improved production titers of cytotoxic metabolites.

- Minor Revisions

Recommend re-reading the manuscript and correcting minor grammatical and syntax mistakes throughout the manuscript.

Response: With the help of an Editor of Applied & Environmental Microbiology, we have carefully re-read the manuscript and corrected some grammatical and syntax errors throughout the manuscript, which are shown in red.

Reviewer 3

Overall this is a scientifically sound manuscript that is thoroughly reported and addresses a question of what do natural products do via the creation of probes to look at downstream physiological stressors with the anasmitocin pathway from *Actinosynnema pretiosum* ATCC 31280. My comments are generally minor and I think the paper should be published with minor revisions, but I believe the manuscript would be improved with the following additions:

1) P9 lines 170-171--It might be helpful to comment briefly on which types of metabolic processes that are essential are disrupted (I know examples are later, but it would improve context for the reader).

Response: As suggested, we have described the necessary metabolic pathways that could be disrupted in the revised manuscript as follows (Line 172-174, Page 9): As most of the AP-3 binding proteins participate in essential metabolic processes, such as cell division, cell-wall assembly, nucleotide biosynthesis and central carbon metabolism.

2) Figure 2--It says the bar is black in the figure legend, but it is actually grey in panels d and e.

Response: We have corrected it in the legend of Figure 2 (Line 213, Page 12).

3) Overall, the text is really small on all the figures, especially Figure 2 which reduces the clarity of the text.

Response: We agree on that the inappropriate of format and clarity of figures caused confusion to readers. Therefore, we enlarged texts in all figures and achieved significant visual improvement.